



# Very severe storm tides in the German Bight (North Sea) and their potential for enhancement.

Iris Grabemann[1], Lidia Gaslikova[1], Tabea Brodhagen[2], and Elisabeth Rudolph[2]

[1]Helmholtz-Zentrum Geesthacht, Max-Planck- Straße 1, 21502 Geesthacht, Germany
[2]Bundesanstalt für Wasserbau, Wedeler Landstraße 157, 22559 Hamburg, Germany

**Correspondence:** Lidia Gaslikova (lidia.gaslikova@hzg.de)

**Abstract.** Storm tides are an essential hazard for the German North Sea coasts. For coastal protection and economic activities, planning information on probability and magnitude of extreme storm tides and their possible future changes is important. This study focuses on the most extreme events and examines whether they could have become more severe under slightly different conditions still remaining within the physical plausibility.

In the face of limited amount of observational data on very severe events, an extensive set of model data is used to extract most extreme storm tide events for locations in the German Bight, in particular Borkum and the Ems estuary. The data set includes water levels and respective atmospheric conditions from a hindcast and future climate realizations without sea level rise describing today's and possible future conditions.

A number of very severe events with water levels exceeding those measured near Borkum since 1906 has been identified in

the data set. A possible further amplification of the highest events is investigated by simulating these events for the North Sea with different phase lags between the astronomical tide given at the open model boundaries and the wind forcing. It was found that superposition of spring tide conditions, different timing of the astronomical high water and atmospheric conditions during the highest storm event would cause an enhancement of the highest water level up to about 50 cm.

The amplified water levels of the two highest events from the data set are used to analyse the effects in the Ems estuary

using a high-resolution model of the German Bight. Additionally, the influence of an extreme river runoff and of sea level rise is studied. The extreme river runoff of $1200\,\mathrm{m^3 s^{-1}}$ increases the highest water levels by several decimeters in the narrow upstream part of the Ems estuary. This effect diminishes downstream. The sea level rise increases the water level in the downstream part of the Ems estuary by the amount applied at the model boundary to the North Sea. In the upstream part, its influence on the water level decreases.

This study may serve as a first step towards an impact assessment for severe storm tides and their implications for coastal areas and activities.



# 1 Introduction

The North Sea (Figure 1) lying at temperate latitudes (51N to 62N) is exposed to the impact of storms mainly occurring from September to April. Storm tides accompanied by severe winds represent a major environmental threat for low-lying coastal areas.

In the modern times, two major storm tide disasters that inflicted heavy losses at the North Sea coasts occurred in the years 1953 and 1962. Since then coastal defenses have been significantly improved throughout the coastline. Mainly due to these measures more recent storms, e.g. 1976 or 2013 caused no severe damages, although higher water levels have been observed locally (e.g. NLWKNa (2010), NLWKNb (2007)). Nevertheless, risk is still present and may increase due to expected climate change. In the course of time, a sea level rise associated with anthropogenic climate change will aggravate the already known problems caused by storm surges. The water levels during storm tides will be higher, the storm tides will reach the inner estuary earlier and the high water levels will last longer. These effects of climate change on storm tides have consequences for coastal protection e.g. for the dike heights or the warning times, but also for such issues as the drainage of low-lying coastal areas.

Coastal protection and adaptation measures usually are a long-term effort. Information about the probabilities of very severe storm tides and their possible changes in the future are needed for planning and design of coastal defenses and protection, for risk assessment and for the assessment on whether or not planned adaptation measures are adequate or robust for a given location. This information is mostly assessed in form of high percentiles obtained from frequency distributions or return value estimates (e.g. Debernard and Røed (2008), Dangendorf et al. (2013), Wolff et al. (2018)). So far, more detailed information and assessment of particular events that are extremely severe and rare are uncommon. Potential sources of such events comprise observations and historical data, modelled data for past, present and future and the events constructed in various ways. The historical data are limited and a priori do not contain severest physically possible events so either they should be modified in a consistent way or additional model data representing e.g. different scenarios or different forecast ensemble members can be additionally used.

The present study aims at identifying and assessing individual extreme events that are highly unlikely but that are still physically possible and plausible and may have extreme consequences. To identify the extreme storm tide events, we initially search through an extensive set of modeled data, which increases the chances to detect the unprecedented events with respect of usage of only historical data. Further, we explore the potential of such events to become even more severe under physically plausible assumptions valid for present day conditions. Finally, we estimate how such events can evolve under hypothetical future climate change conditions.

There are several key processes determining water level increase during a storm and their modification may lead to enhancement of identified extreme events. Among others, variations in the atmospheric conditions leading to changes of storm track and/or intensity over sea may entail alterations in the storm tides near the coast. In particular, in the project MUSE Jensen et al. (2006) took a dynamic ensemble approach. They analyzed the extent to which enhancement of observed storm tides could be caused by various atmospheric developments of observed storms. The atmospheric variations in this case were represented by different timing of initial conditions used for the atmospheric forecast and corresponding ensemble simulations. Another study





held within the project XtremRisK (Gönnert et al. (2013), Oumeraci et al. (2015)) developed a more combinatorial approach merging estimates of various storm tide components such as surge, external surge, tides and their non-linear interactions derived from observation. Both studies were mainly focused on the Elbe estuary and resulted in constructing and investigating
events exceeding the observed ones.

Without changing the atmospheric forcing possible amplification can occur due to different configurations of existing atmospheric situations and astronomical tide. In particular, altered timing of atmospheric storm relative to the tidal phase may lead to variations in maximum water level. In addition to semidiurnal tidal variations the longer fluctuations of the tidal components can be considered reflecting the situation where particular atmospheric storm may coincide with spring tide instead of neap
tide. In the present study we pursue this strategy to investigate the potential for very severe storm tides to be enhanced.

Whereas Jensen et al. (2006) looked at particular observed storms and the amplification of their peak water levels, the current study deals with a large set of met-ocean hindcast and climate realizations to detect extraordinary storm events, focusing on both storm tide height and duration. The used climate realizations incorporating CMIP3 and CMIP5 scenarios reflect only the changes in the atmospheric conditions and do not include mean sea level rise and local bathymetry changes. A variety of future
climate realizations underlines large uncertainties regarding possible future changes in storm climate for the region of interest (e.g. Feser et al. (2015), Ganske et al. (2016)). Hereafter we assume that extremes from the used simulations represent also plausible events for the present climate conditions as storm statistics in these simulations show no or minor significant changes towards 2100 in combination with very strong inter-decadal variability for wind speed and surge levels (e.g. Gaslikova et al. (2013)).

From this data set the most extreme storm tide events were selected for three distinct parts of the German Bight - East Frisian and North Frisian coasts and Elbe mouth (Figure 1). A set of dynamical large-scale simulations was produced to examine whether the identified storm tides could have become more extreme under different constellations of peak winds and tides. Hereby, a regional hydrodynamic model, which covers the North Sea and parts of the North East Atlantic to ensure the incorporation of external surges was used.

To investigate local effects of such extremely severe events near the coast and specifically in the estuaries, the Ems estuary was chosen for further experiments and analyses. The estuary represent one of the main German estuaries. In addition to dykes along the North Sea coast and the whole estuary the upper Ems estuary is protected by a storm surge barrier. Operating the barrier influences the water levels both upstream and downstream of the barrier (Rego et al. (2011), BAW (2007)). Such effects under extreme storm tide contidions are of additional interest. Moreover, the town Emden as an example for a typical harbor
town with importance for marine trade, was chosen as a focus point within the estuary. To adequately transfer the acquired extreme storm tides to the coasts and assess their impact within an estuary, a more detailed hydrodynamic model for the German Bight including the German estuaries has been used (Figure 1). Additional factors, which may lead to the amplification of water levels at the coast and which are more relevant at local scales and shallow water (effects of varying river discharge and possible future sea level rise) were considered and incorporated in the sensitivity study here.





## 2 Data, methods and experiments


### 2.1 Hydrographic properties

The south-eastern and north-western coasts of the German Bight (Figure 1) are mainly endangered by storm winds from westerly to northwesterly and by southwesterly to westerly directions, respectively. The tidal wave propagates anti-clockwise from the East-Frisian to the North-Frisian coast. Due to the funnel-shaped German Bight the mean tidal range increases from

about 2.4 m near Borkum near the outer border of Ems estuary to about 3 m in the outer Elbe estuary and decreases to about 2.6 m near Amrum (e.g. DGJa (2014)). In outer parts of the estuaries of Ems, Weser and Elbe the mean tidal range can exceed 3 m (e.g. Niemeyer and Kaiser (1999)). Thus, a specific storm in the southern North Sea has different influences on the water levels at the different coastal strips and in the estuaries.

The Ems estuary is situated in the German Bight in the southern North Sea at the border between the Netherlands and

Germany (Figure 1). Coming from the wide mouth of the estuary near the island of Borkum it is narrowing towards Knock, but again widening into the Dollart bay south of Emden. Upstream of the Dollart the narrow and shallower part of the Ems estuary begins. The influence of the tide can be observed until Herbrum. At the mouth of the Ems near Borkum the tide is characterised by mean tidal high water MHW NHN + 1,15 m and mean tidal low water MLW NHN - 1,31 m (DGJa (2014), NHN (Normalhöhennull) presents the German standard elevation zero.) In the center of the estuary at Emden the mean tidal

range increases to 3,28 m with mean tidal high water MHW = NHN + 1,48 m and mean tidal low water MLW = NHN - 1,80 m (DGJb (2018)). The mean freshwater discharge into the Ems estuary is 80 $m^3s^{-1}$, the highest discharge observed is 1200 $m^3s^{-1}$ (February 1946) (DGJb (2018)). Large freshwater discharges occur frequently in the months from January to April (Krebs and Weilbeer (2008)).

### 2.2 Data set

For the detection and ranking of extreme storm tides, a set of numerical simulations has been used for which atmospheric as well as marine data exist and for which the water levels were simulated with the same hydrodynamic model. This set includes a multi-decadal hindcast (Weisse et al. (2014), Weisse et al. (2015)) based on downscaled NCEP-NCAR global reanalyses (Kalnay et al. (1996)) and six multi-decadal climate change realizations up to 2100 with respective control simulations. The atmospheric simulations include four realizations of the CMIP3 emission scenarios A1B and B1 and two realizations of the

CMIP5 emission scenario RCP8.5 (for emission scenarios see e.g. Nakicenovic and Swart (2000), Houghton et al. (2001) and Stocker et al. (2013), for CMIP5 simulations see e.g. Taylor et al. (2010)). They were simulated with different global models (ECHAM5-MPIOM (e.g. Röckner et al. (2003), Marsland et al. (2003)), EC-EARTH (e.g. Hazeleger and Coauthors (2010)), CMCC (Scoccimarro et al. (2011))) starting at different initial conditions. The global atmospheric simulations were downscaled with different regional circulation models (different versions of CCLM (e.g. Rockel et al. (2008), Hollweg et al.

(2008)), RCA4 (e.g. Samuelsson et al. (2011))) before they were used to force the hydrodynamic model TRIM-NP (Kapitza and Eppel (2000), Pätsch et al. (2017)) to calculate water levels in the North Sea and Northeast Atlantic (e.g. Gaslikova et al. (2013), see also Figure 2).





The climate realizations do not include any rise in mean sea level. Water level changes are due to changes in the atmospheric forcing only. Furthermore, possible changes in bathymetry within the course of the time are neglected in the hindcast as well
as in the climate realizations.

## 2.3   Selection of events and enhancement experiments

The analysis of extreme storm tides is mainly focused on the East-Frisian coast in particular on Borkum and the Ems estuary. However, the impact of storms in the North Sea varies along the coasts depending on the wind direction and the resulting wind set up. Therefore, from the data set, time series of hourly water levels were extracted for a location seaward of the island of
Borkum (in the following mentioned as "Borkum") and two other locations in the German Bight (Figure 1): one location in the outer Elbe estuary (mentioned as "Elbe Mouth") and one location seaward of the North-Frisian island of Amrum (mentioned as "Amrum").

Figure 2 describes the workflow for the simulation of the original water levels included in the data set and for the construction of the amplified water levels. A potential amplification due to tidal variations is tested for selected events at Borkum, whereas
Elbe Mouth and Amrum are used to compare the effects at Borkum with those at other coasts of the German Bight. A potential amplification of the selected events in the North Sea as well as nearer to the coast in the German Bight and the Ems estuary is investigated in the following by four steps.

In step 1, extreme storm events are selected from the corresponding time series using three criteria:

-  height of water levels,

-  duration of water levels countinuously exceeding 1.15 m (MHW at Borkum, DGJa (2014)) and

-  chain of events within one week.

Water levels are considered with respect to NHN. The selected events for Borkum are ranked with respect to their water levels and their durations. For the further analysis of a possible amplification, the highest event, the longest event and the strongest event chain from the selected events were chosen. In the following these events are mentioned as "EH", 'EL" and
"EC", respectively.

In step 2, possible amplification of the selected extreme events due to different combinations between wind field and astronomical tide was tested. Maximum water levels may be increased by variations of relative propagation and arrival time of tidal high water and atmospheric storm. They may also become higher if the specific storm occurs around spring tides rather than around neap tides.

Thus, ensembles of large-scale North Sea water level simulations for each selected event were generated. For ensemble one, the astronomical tide given at the open model boundaries was shifted hourly within +/-6 h around the wind speed maximum. For ensemble two, the highest spring tide found in the respective climate realization was used instead of the original tide and the astronomical tides were shifted again hourly. For each member of ensemble one and two water level time series were extracted for the three locations, in these cases with a time resolution of 20 minutes. The time series for Borkum were analysed with





respect to the strongest amplification. Furthermore, the effects of the amplification procedures for Borkum were compared to the corresponding effects at Elbe Mouth and Amrum.

Respective data from the ensemble members with the highest amplified water levels near Borkum (in the following identified by "_a") for each event were used for further fine-grid simulations of the German Bight and the Ems estuary in steps 3 and 4.

In step 3, high resolution water level simulations for the German Bight and the attached estuaries for the ensemble member

with the highest amplified water levels near Borkum for the selected events derived from step 2 were performed.

In step 4, the events from step 3 were further amplified by applying an increased river runoff to examine the impact of runoff variations and a sea level rise to place the results in the context of future climate change. For these amplification simulations the highest observed river runoff for the Ems of $1200 \, \mathrm{m^3 s^{-1}}$ (1946, DGJb (2018)) was assumed. This extreme river runoff was measured in February 1946, i.e. in a season where storm tides are probable. Furthermore, simulations with two sea level rise

scenarios of 50 cm and of 100 cm were investigated. These values cover the likely range of median values for the global sea level rise as well as the bandwidth of the local sea level rise for the North Sea until 2100 as reported by Stocker et al. (2013). The sea level rise was applied at the boundary of the German Bight model to the North Sea by shifting the boundary values for water level by the selected amount of sea level rise.

In order to investigate the impact of the storm surge barrier in the Ems on water levels, the storm tides are simulated with

open and with operated barrier in steps 3 and 4.

## 2.4 "North Sea" model for the large-scale simulations

For consistency reasons, the simulations in steps 2 were performed with the same hydrodynamic model TRIM-NP which was used previously for the considered data set of hindcast and climate realizations. The model TRIM-NP (Nested and Parallized, Kapitza and Eppel (2000), Pätsch et al. (2017)) is based on the model TRIM (Tidal Residual Intertidal Mudflat) developed

by Casulli and Cattani (1994)) and was used in 2D mode. The model domain covers the North Sea and adjacent parts of the Northeast Atlantic (Figure 1) to allow the generation of realistic external surges. The model solves the Reynolds-averaged Navier Stokes equations on a regular Arakawa-C grid with Cartesian coordinates and is used in the present study with a resolution of 12.8 km x 12.8 km without further nesting. The model time step was 4 minutes and the output was stored every 20 minutes. Drying and wetting of near-shore points is enabled. The water level simulations were driven by the 10 m height

wind and mean sea level pressure (SLP) fields from the atmospheric data mentioned above and by astronomical tides from the FES atlas (Lyard et al. (2006)) at the lateral open boundaries. The wind influence is parameterized using an approach from Smith and Banke (Smith and Banke (1975)). While this approach is based on wind speed measurements between 3 and $21 \, \mathrm{ms^{-1}}$ which are exceeded during storm surges, previous studies have shown that this approach is suitable for the North Sea and applicable for storm surges (Jensen et al. (2006)). For a detailed description of the original water level simulations and

model performance see Gaslikova et al. (2013) and Weisse et al. (2014).





## 2.5 "German Bight" model for the fine-grid simulations

For the high-resolution modelling of the German Bight and the attached estuaries of the rivers Ems, Weser and Elbe (Figure 1) for the selected events in steps 3 and 4, the hydrodynamic numerical model UnTRIM$^2$ (Casulli (2008)) is used.

UnTRIM$^2$ is a 3D finite difference / finite volume numerical model. It solves the shallow water equations and the transport equation of salt on an unstructured orthogonal grid (Casulli and Walters (2000)). The use of the subgrid technology described by Casulli (2008) allows discretizing the model bathymetry with a much finer resolution than the computational grid. In areas like the German Wadden Sea with its large tidal flats, this allows describing dry and wet areas in greater detail as well as better representation of the water volume. Thus, the bathymetry can be captured in detail while the computations can still be performed on a relatively coarse grid. As a result, large time steps can be used and the computational costs are kept low. The algorithm also guarantees conservation of mass and water depths greater than zero regardless of time step size and is unconditionally stable.

The German Bight model is forced by wind, river runoff, salinity and water level. For these simulations, the same wind fields which were used in step 2 for the North Sea simulations were used. In UnTRIM$^2$, the wind parameterization is similar to that in TRIM-NP. River runoff is applied at the upstream end of the estuaries. For the Ems, a constant runoff of $80 \, \mathrm{m}^3 \mathrm{s}^{-1}$ (average for 1942-1915,DGJb (2018)) was used. Water level and salinity are applied at the open boundary towards the North Sea (Figure 1). Water levels were derived from the North Sea simulations with TRIM-NP (see step 2). A constant salinity of $33 \, \mathrm{psu}$ was used which is a common value for that region of the North Sea (BSH (2016)).

The storm surge barrier (Figure 1) is included in the subgrid topography of the model and can be operated at run time. Based on the balance between coastal protection and nature conservation the barrier should protect the estuary against storm tides higher than NHN + 3.70 m. The barrier is closed when water levels at the barrier are exceeding NHN + 3.50 m and it is reopened when water levels upstream and downstream of the barrier are equal. In order to ensure the protective function of the storm surge barrier in case of a sea level rise of 100 cm, the height of the gates were increased from its original 7 m to 8 m in nature to 9 m in the model.

## 3 Results

### 3.1 Extreme storm tides at the coasts of the German Bight

#### 3.1.1 Selected extreme storm tides for Borkum

Different classifications of storm tides exist using e.g. water levels above a reference height or the probability of water levels. Here, the classification of the Bundesamt für Seeschifffahrt und Hydrographie (Federal Maritime and Hydrographic Agency, see Müller-Navarra et al. (2003)) is used: A storm tide is an event with water levels exceeding mean tidal high water (MHW) at least by 1.5 m, a severe and a very severe storm tide denote events exceeding MHW by 2.5 m and 3.5 m, respectively.





Following the procedure described in the previous chapter, events were selected for Borkum and ranked with respect to their water levels and their durations. In Figure 3 time series for the five highest storm tides extracted from the data set are compared with the highest observed storm tides for Borkum showing that the data set includes storm tides higher than observed during the past 110 years. The events observed in 2006 and 2013 denote the second and third highest storm tides (DGJa (2014)),

measured at the tide gauge since the beginning of the 20th century. The highest observed storm tide with 4.06 m occurred in 1906 (NLWKNa (2010)). The five original simulated events are about 40 to 65 cm higher than the event observed in 1906.

The highest event (EH, Figure 4 top and red curve in Figure 3) with a maximum high water of 4.73 m describes a very severe storm tide and was found in one of the B1 realizations (February 2030, for detailed description of the realizations see Gaslikova et al. (2013)). This event has also a comparably long lasting time period with water levels higher than the long-term MHW of

1.15 m (DGJa (2014)).

The chosen chain of storm tides (EC, Figure 4 bottom) was found in one of the A1B realizations (November 2030). The longest event (EL) with water levels exceeding MHW for 45 h is included as first event in the chain of storms. Furthermore, the highest high water of 4.66 m in EL just reaches the water level for a very severe storm tide and presents the second highest event extracted from the data set (orange curve in Figure 3). EL/EC includes in total seven storm high waters within less than

eight days.

The effective wind is used here as a relevant representative of the local wind activity. It is the projection of the horizontal wind vector on that direction which is most effective in producing surges at the coast (see e.g. Ganske et al. (2018)). During EH and EL/EC the single events follow the effective wind variations shown exemplarily for Borkum (dashed black curves in Figure 4). According to the classification of general weather situations causing severe storm surges along the German coasts

(e.g. Kruhl (1978)) the storm tracks causing events EH and EL (not shown here) belong to the "North-West Type" (for areas of tracks of the different categories see Figure 3 in Gerber et al. (2016)) .

### 3.1.2 Amplification analysis for selected extreme storm tides for Borkum

In the original event EH the maximum high water coincides with the maximum of the effective wind and the maximum surge occurs about four hours before the astronomical high water. Figure 5 displays the original event EH and the ensemble member

with the highest high water obtained from the experiments with shifting wind and astronomical tide hourly against each other. In this case, a 5 h shifting leads to the highest water levels. The event EH consists of two high waters (peak 1 and peak 2) classified as at least severe storm tides. After the amplification the highest peak 2 becomes smaller whereas the lower peak 1 originally reaching 3.93 m now comes up to 4.88 m. Due to the diurnal inequality the astronomical tide underlying peak 1 is about 20 cm higher than that underlying peak 2 and due to the 5 h shifting it coincides with stronger wind velocities, whereas

the astronomical tide of peak 2 coincides with weaker wind velocites. Thus, by only shifting the astronomical tide against the wind field, an amplification of the highest high water in the event EH of 15 cm is obtained.

Figure 6 shows the ensemble member with the highest high water from the simulation experiments with replacement of the original astronomical tide by the largest spring tide together with hourly shifting between astronomical tide and wind field. The





high water of the replaced astronomical spring tide is about 40 cm higher than the astronomical high water of peak 2 in event

EH. This amplification procedure results in a high water of 5.23 m presenting an amplification of 50 cm.

For the original event EH the time period with water levels greater than MHW is about 33 h corresponding to approximately three tidal cycles. Due to the amplification procedures this time period varies up to +/- 1 h except two ensemble members for which it is prolonged up to about four tidal cycles. For these two members which show no amplification concerning the highest high water the low water before peak 1 and the low water after peak 2 do not fall below MHW. For all other members either

the low water before peak 1 or the low water after peak 2 falls below MHW.

In case of the event chain (EC) including the longest event (EL, Figure 4), both amplification procedures - shifting of the astronomical tide against the wind and replacement of the original astronomical tide with the highest spring tide together with shifting - results in an increase of the highest high water by only few centimeters. In the original event EL the highest high water already coincides with an astronomical spring tide only few centimeters lower than the highest one. Thus, both applied

procedures lead to relative changes of the three highest water level peaks, however not to a substantial absolute increase of the highest water level during EL. Furthermore, the length of EL shows nearly no changes. Possible amplification was tested for EL; for the two following smaller events in EC there are ensemble members showing an increase of single high waters up to 20 to 30 cm.

For further analysis of effects in the Ems estuary, the original EL/EC water levels are used whereas for event EH the amplified

water levels due to the spring tide replacement together with tide shifting are used, in the following mentioned as EH_a (Figure 6, red curve).

### 3.1.3   Comparison of amplified extreme storm tides at different coastal strips

EH and EL/EC (Figure 4) are analyzed for Borkum for possible amplification. Although these highest events are selected and ranked for Borkum, they cause severe storms at the other coasts of the German Bight represented here by Elbe Mouth and

Amrum. In particular, EH and EL/EC give the second and third highest events at Elbe Mouth and the third and fourth highest events at Amrum in the data set. From Figure 4 it can be seen that the specific ranking of the single high waters during each event differs between the locations, but the duration of the events is comparable. The high water occurs about 1.3 h and 2 h later at Elbe Mouth and Amrum, respectively, compared to Borkum.

The effects of the amplification procedures adjusted for Borkum are exemplarily compared to those at Elbe Mouth and

Amrum for event EH (Figure 6). In general, the water level changes caused by the amplification procedures for Borkum are similar at the other two locations. In case of the 5 h shift of the astronomical tide, peak 1 increases for Elbe Mouth and Amrum as well. In case of the replacement of the original astronomical tide with the spring tide and hourly shifting, peak 1 shows no or only small changes whereas peak 2 increases. Nevertheless, for location Elbe Mouth in the outer Elbe estuary and for Amrum at the North-Frisian coast the relative impact of the two procedures differs from that for Borkum. At Elbe Mouth

both procedures cause similar maximum high waters of 5.35 m (+ 49 cm) and 5.23 m (+ 37 cm), respectively, during the event, whereas at Amrum the 5 h shifting results in the highest high water of 5.25 m (+ 56 cm) as there the original peak 1 is higher than peak 2.





The particular amplification mechanisms were adjusted to maximize water levels at Borkum. Thus, other time lags might lead to higher water levels at Elbe Mouth and Amrum. This is demonstrated by the olive curves in Figure 6 which show the

highest amplified water levels for these two locations for differing ensemble members. For Amrum the blue and olive curves reach the same highest high waters, but for the olive curve the amplification is based on peak 2. For Elbe Mouth the olive curve provides an amplification of 72 cm. The olive curves both for Elbe Mouth and Amrum originate for the same ensemble member which incorporates replacement by the spring tide together with tide shifting.

As for Borkum also Elbe Mouth and Amrum show some changes in the time period with water levels exceeding MHW.

For Elbe Mouth, this time period is reduced by about one tidal cycle for few members mainly with replaced spring tide. For Amrum, this time period is prolonged up to about one tidal cycle for few ensemble members.

### 3.2 Extreme storm tides in the Ems Estuary

#### 3.2.1 Impact of Q and SLR on water levels at Emden

Based on the fine-grid simulations of the German Bight, the impact of additional amplifications on the selected extreme events

EH_a and EL/LC is investigated for the Ems estuary. Here, additional amplification refers to a sea level rise (SLR) and to an increase in river runoff (Q) of the Ems.

Time series of the water levels at Emden in the Ems estuary are shown in Figure 7 for event EH_a and in Figure 8 for event EL/EC with operated storm surge barrier for a simulation without amplification and for simulations with increased Q and applied SLR. EH_a reaches peak water levels of 6.61 m at Emden without additional amplification which is 5.13 m higher than

the long-term mean tidal high water level MHW of 1.48 m (DGJb (2018)) and leads to the classification of EH as a very severe storm surge which is in agreement with its classification at Borkum. EL/EC reaches peak water levels of 5.96 m at Emden which also classifies the event as a very severe storm surge. Both events reach water levels that exceed the highest observed water level of 5.17 m at Emden (1906, DGJb (2018)).

Changing the river runoff from $80\,\mathrm{m^3s^{-1}}$ to $1200\,\mathrm{m^3s^{-1}}$ increases the tidal high and low waters at Emden only by a few

centimeters (see Figures 7 and 8, red and dashed grey lines). This effect is even weaker for the storm tides (see events with open storm surge barrier in Table 1). In the wide and deep estuarine part near Emden the tidal volume strongly exceeds the river runoff so that the impact of river runoff on water levels is small. As the tidal volume is increased during the storm surge, the impact of river runoff is even smaller during this period.

At Emden, applying a SLR to the events leads to an increase in tidal high water, tidal low water and the highest water level

during storm tide in the range of the applied SLR (Figures 7 and 8 and Table 1). This behaviour can be seen in both EH_a and EL/EC. The observed influence of river runoff and sea level rise agrees with the behaviour analyzed in a sensitivity study by Rudolph (2014).

Increasing the river runoff results at Emden in nearly no change in the occurence time of the highest water during storm surge (see events with open storm surge barrier in Table 1). The increased water depth caused by a sea level rise increases the





propagation velocity of the tidal wave entering the Ems estuary which causes the tidal high water to occur earlier by 10 to 20 minutes at Emden (Table 1) for the events investigated.

For the event EH_a the time period with water levels greater than MHW is about 33 h similar to the time period at Borkum. Due to an increase in runoff to $1200\,\mathrm{m^3s^{-1}}$ or to a sea level rise of 0.5 m this time period shows only small changes less than an hour. But for a sea level rise of 1 m this time period is prolonged by one tidal cycle up to about 45 h (Figure 7). For event EL the time period with water levels continuously greater than MHW is about 2 tidal cycles. This differs from the conditions at Borkum where four consecutive tidal cycles are continously above MHW. In the Ems estuary the tidal range is greater than at Borkum with lower low waters and higher high waters. Thus, the time periods around the two low waters following the highest high water are below MHW for about 2 to 3 hours. In case of a sea level rise of 1 m the event EL is prolonged by two tidal cycles as the mentioned two low waters become higher than MHW (Figure 8).

### 3.2.2 Impact of Q and SLR on highest water levels along the Ems estuary

To investigate the influence of Q and SLR along the Ems estuary, the highest water levels HW during EH_a and EL at each location along the longitudinal profile are analyzed for simulations with an open storm surge barrier. Closing the barrier separates the estuary into two parts and alters the effects of Q and SLR.

Figure 9 shows the impact of an increase of Q from $80\,\mathrm{m^3s^{-1}}$ to $1200\,\mathrm{m^3s^{-1}}$ on HW for both EH_a (black lines) and EL/EC (red lines). For both events, the increased river runoff rises the highest water levels by several decimeters in the narrow and shallow upper part of the Ems estuary upstream of the Dollart (bight in the Ems estuary south of Emden, Figure 1). The influence of Q on the highest water levels decreases towards Dukegat where the Ems becomes deeper and wider and disappears towards the mouth of the estuary. As mentioned before, Emden is located in an area of the Ems estuary where the influence of the river runoff on the highest water levels is in the range of some centimeters (see also Figures 7 and 8 and Table 1).

Upstream of Papenburg the influence of the bathymetry on the highest water levels during storm surge can be observed clearly marked by a sudden decrease in HW in case of low discharge. In the area of Papenburg the estuary is very narrow, the dike line is close to the estuary, whereas the upper part of the estuary is characterized by wide foreshore areas that are flooded only during events of high discharge or storm surges. In addition the depth of the estuary is decreasing significantly upstream of Papenburg, as seagoing ships are not using this part of the Ems estuary.

The described bottleneck close to Papenburg prevents the water during storm surge to enter the upstream area undisturbed, which results in lower water levels in this area. For events with high river runoff the wide foreshore areas upstream of Papenburg are already flooded before the storm surge depending on the amount runoff and the height of the tide before the storm tide.

Increasing the SLR from 0 to 1 m while the river runoff remains unchanged at $80\,\mathrm{m^3s^{-1}}$ leads to a longitudinally varying increase of the highest water levels along the whole estuary for both EH_a and EL/EC (Figure 10 top). The difference in maximum water levels between a simulation with SLR and a simulation without SLR (Figure 10 bottom) shows for both EH_a and EL/EC that the maximum water levels downstream of Leerort are increased by the amount of the applied SLR with small deviations in the range of a few centimeters. The impact of the SLR on maximum water levels decreases between Leerort and


Papenburg. The rate of decrease depends on the magnitude of the applied SLR. For a SLR of 0.5 m and 1 m maximum water levels drop by about 20 %. Upstream of Papenburg, the impact of the SLR changes depending on the event (Figure 10).

In the lowlands close to the mouth of the Ems draining of urban (e.g. Emden) and agricultural areas (e.g. Knock) is of major interest. The aim of the sewer at Knock is to drain the low lying hinterland (with a ground level of about NHN + 0 m) and keep the inland water level at Knock lower than NHN - 1.40 m (KLEVER (2018)). At Knock the mean low water MLW is NHN - 1.58 m so that draining without pumping is only possible for a short time even during mean tides. Caused by long lasting high water levels during storm tides only restricted draining is possible.

For the chain of storm tides EC (see Figure 8) even without amplification pumping is needed nearly during the whole period of 176 hours (see Table 2). The water must be pumped against a water level in the Ems higher than MHW for about 90 hours. This period will increase by about 40 hours in case of a sea level rise of 100 cm.

### 3.2.3  Influence of the storm surge barrier in the Ems estuary

For the investigation of the impact of SLR and runoff along the Ems estuary as shown above, the storm surge barrier in the
Ems is considered to be open. When operated, the storm surge barrier in the Ems has a significant influence on highest water levels both upstream and downstream (Figure 11). The barrier is closed at a defined water level of NHN + 3,50 m and reopened when the water levels on both sides of the barrier are equal.

In the protected area upstream of the barrier the water levels are no longer influenced by the storm surge coming from the North Sea. Only the amount of river runoff that flows into the protected area in the period the barrier is closed contributes to
the highest water level.

Downstream of the barrier, highest water levels increase compared to a simulation with an open barrier (black curves in Figures 9 and 11, see also time series at Emden in Figures 7 and 8, red and dashed black curves). This is due to two main reasons: Firstly, the sudden stopping of the impulse of the tidal wave at closure leads to a positive surge downstream of the barrier and a negative surge upstream of the barrier. The positive surge induces a self-oscillation in the Dollart basin in which
the period of oscillation is depending on the geometry and the actual water depth of the Dollart basin. The created surge will be weaker when current velocities are lower and can be avoided when closing the barrier at slack water time.

This effect can be observed e.g. in Figure 7 looking at the water level of the event EH_a (red line). The storm surge barrier is closed at a defined water level (NHN + 3,50 m) and not at a defined time in the tidal phase. For the first storm tide the storm surge barrier is closed during flood current. The induced surge and the subsequent oscillation causes an unsteady rise of the
water level. For the second storm tide the water level of 3,50 m is reached during slack water time. As the storm surge barrier is closed during a period of nearly no current velocity, no surge is induced and a steady rise of the water level is observed. This behaviour was investigated in detail in BAW (2007).

The second process increasing the highest water levels is the shortening of the estuary that takes place when the barrier is closed. This reduces the stretch where dissipation of the tidal wave can occur and leads to the reflection of a more energetic
wave and thus increase the highest water levels. This behaviour has been investigated in other studies and is also described in BAW (2007). The amount of the increase in highest water levels has been studied in e.g. Rego et al. (2011).





Summarizing, a closed storm surge barrier will always lead to increased highest water levels downstream of the barrier but the magnitude of the increase depends on the current velocity conditions present at closure. In case of the analysed events, at Emden this increase ranges between 19 and 25 cm (Table 1). In general, the highest water levels are reached close to the storm

surge barrier, they decrease towards the river mouth. Closing the barrier keeps the storm tide out of the area upstream. Only the river runoff fills the protected area during the period when the barrier is closed. Consequently, closing the barrier during a storm tide leads to significantly lower highest water levels upstream of the barrier (Figure 11).

Applying a SLR and an increased Q to events with operated storm surge barrier leads to increased highest water levels downstream of the barrier due to the SLR and increased highest water levels upstream of the barrier due to the runoff coming

from upstream (see Figure 11). This holds true for all events and respective simulations. Figure 11 demonstrates how the water level upstream depends on the length of period with closed barrier. For a SLR of 1 m and a runoff of $1200 \, \text{m}^3\text{s}^{-1}$, the water levels during event EH_a are considerably higher than those for event EL although for the reference cases (no SLR, mean runoff) the water levels are similar. For event EH_a the barrier has to be closed for the first storm tide (see Figure 7) and could only be reopened after the second storm tide due to the considerably elevated water levels around low tide. This leads

to a continuous closure period of 17 h. For event EL the water levels allow to close and open the barrier for each storm tide separately (see Figure 8) leading to a closure period of 7 h 5 min. Combining these different closure periods with the extreme runoff results in lower water levels for EL than for EH_a upstream of the barrier. In case of mean runoff and no sea level rise the length of the closure period is not influencing the highest water level during the storm tide upstream of the barrier. It shows that the protected area upstream of the barrier is big enough to store even the extreme discharge of $1200 \, \text{m}^3\text{s}^{-1}$ for all closure

periods investigated. For all events and amplifications the highest water levels upstream of the operated barrier remain lower than those reached in case of the open barrier (see Figure 10 and Figure 11).

The highest water levels at Emden for the simulations without further amplification and an operated storm surge barrier are 6.61 m for EH_a and 5.96 m for EL/EC (Table 1). Applying amplified conditions (Q=$1200 \, \text{m}^3\text{s}^{-1}$ and SLR=1 m) leads to an increase of highest water levels to 7.65 m for EH_a and 7.01 m for EL/EC (Figures 7 and 8). The highest measured water level

at Emden is 5.17 m (DGJb (2018)). Thus, the extreme events EH_a and EL/EC identified and elaborated in this study exceed this water level even without the application of further amplification through river runoff and sea level rise.

## 4   Summary and discussion

This study aims to find extreme storm tides in the North Sea and Ems estuary that are physically possible but have not been observed yet. Numerical simulation data from both hindcast and climate realizations have been searched to detect extreme

storm tides, i.e. storms causing either very high water levels (event EH) or water levels exceeding mean tidal high water for a longer time (event EL) or where multiple storm tides occur within one week (event EC). Both extreme events (EC contains EL) selected according to these criteria originate from the climate realizations and there from the first half of the emission scenario period, although from two different realizations. This underlines the strong inter-decadal variability and the absence





of a considerable increase of extreme storm tides towards 2100. Thus, the found highest water levels exceeding the water levels

measured since the beginning of the 20th century at Borkum (Figure 3) could be possible under present-day conditions.

Using numerical simulations for the North Sea, the selected events were amplified by shifting the astronomical tide against the wind field for optimization of their interaction and by inserting the highest spring tide from the data set. By these amplification procedures based only on the co-timing of the atmospheric storm and the tidal phase, the water level at Borkum is increased by about 50 cm and a maximum water level of 5.23 m is reached for the event EH_a, thus, exceeding the highest

measured event in 1906 by more than 1 m (see dashed red line in Figure 3). Moreover, the enhancement mechanisms proposed in this study except sea level rise are realistic under the present day conditions and thus a storm tide like the amplified EH_a event could occur nowadays.

Using a high-resolution model for the German Bight and the Ems estuary, the extreme events EH_a and EL/EC were further studied in the context of an extreme river runoff of $1200 \, \mathrm{m^3 s^{-1}}$ and increased mean sea level by 0.5 m and 1 m. The river runoff

has the largest impact on highest water levels upstream of Herbrum in the narrow part of the Ems where it leads to an increase of about 1 m. The impact decreases downstream as the Ems becomes wider and deeper and disappears completely downstream of Dukegat. The amplified conditions in the North Sea due to sea level rise increase the water levels in the estuary from the mouth up to the area of Papenburg by approximately the applied amount of sea level rise. Upstream of Papenburg, the river runoff dominates and the influence of sea level rise on highest water levels decreases. Both the sea level rise and the increase

in river runoff lead to an increase in water levels and to a longer duration of higher water levels along the Ems estuary. In addition, a sea level rise results in an earlier occurrence of the highest water level during storm surge in the central part of the estuary in the order of 10 to 20 minutes (Table 1).

Against the background of climate change, coastal protection strategies and usage of the hinterlands it is not only important to know the possible height of an extreme event but also its duration. Moreover, the event EC shows that several high storm

tides within a week could be possible. The low-lying land protected by dykes in this area is drained both using the gradient in the water level towards the Ems and with pumps. A prolongation of the duration of higher water levels in the Ems will hinder the natural drainage. The infrastructure in terms of more powerful pumps must be improved because the water has to be pumped for a longer period against higher water levels in the Ems estuary.

In the Ems estuary at Emden, the highest water level for the event EH_a is 6.61 m with operated storm surge barrier and

without further enhancement. In case of a runoff of $1200 \, \mathrm{m^3 s^{-1}}$ and a sea level rise of 1 m it reaches 7.65 m. These water levels exceed the highest water level observed in the event in 1906 by about 1.4 m and 2.4 m, respectively. Nevertheless, the simulated highest water levels as listed in Table 1 do not reach today's dyke height at Emden of NHN + 7,60 m except for two cases which include a future sea level rise of 1 m. The upper part of the Ems estuary is protected by the storm surge barrier even against extreme events with amplified discharge or sea level rise.

The obtained amplified water levels for event EH_a of 5.23 m at Borkum and of 6.61 m at Emden are in a similar order of magnitude as the maximum water levels of 4.99 m and 6.09 m, respectively, reported by Jensen et al. (2006) (see there Table 10 for an ensemble member of 1976) as an estimate of extreme event with low probability of occurrence. There, the investigation was focused on the Elbe estuary. Possibly, other ensemble members than reported might result in higher water





levels for Borkum and Emden. Still the comparability of extreme water levels estimated by different procedures and based on
different original data sets supports the plausibility of the results. Moreover, there is a potential for further enhanced realistic
storm tide events to emerge when both methods, namely varations in atmospheric conditions as done by Jensen et al. (2006)
and interplay with different tidal phases as done in the present study, are combined.

Depending on track, intensity and velocity, each storm affects the German coastal stripes differently. For the East-Frisian
coast storm winds from northern directions lead to higher storm tides whereas for the North-Frisian coast storm winds from
western directions have more impact. Thus, the ranking of extreme storm tide events elaborated in this study differs in detail
for the different coastal stripes of the German Bight. As this work focuses on the East-Frisian coast with the Ems estuary, the
amplification procedures were adjusted specifically for Borkum. However, the methods for the identification and amplification
of storm tides used here could be transferred to other coasts and estuaries.

So far, a fixed bathymetry was assumed for all simulations. However, the heterogeneous bathymetry of the German Bight, in
particar the Wadden Sea and the estuaries, has been subject to changes due to natural processes and anthropogenic influences
which will proceed in the future. Consideration of changing bathymetry would give an insight on the effect of morphodynamic
states on extreme storm tides.

In the present study the effects of a coincidence of a severe storm tide and extreme runoff were assessed to give an upper limit
of water levels. So far, an independent probability of occurrence of extremes was assumed. Consideration of joint probabilities
or consideration of them as a compound event might narrow down the range of possible water level extremes.

Events like event EC with a series of storm tides within a week might require special arrangementss for the management
of the impact of the storm tide. Not only the drainage of the hinterland must be sufficient, but also manpower to watch and
operate coastal protection measures must be avaiable in adequate numbers. The drainage situation may become worse in case
of the coincidence of a storm event with heavy rain. The results of this study may contribute to the development of a flexible
adaptation route to the impacts of climate change in coastal areas considering the interests of coastal protection, draining of
the hinterland and navigation in the waterway Ems.

*Author contributions.* The simulations for the North Sea (1) and the Ems estuary (2) have been performed and analysed by IG and LG (1)
and TB and ER (2), respectively. All authors contributed to the preparation of the manuscript.

*Competing interests.* The authors declare that they have no conflict of interest.

*Acknowledgements.* The observational data for Borkum were kindly provied by the German Federal Maritime and Hydrographic Agency
(BSH) and German Federal Waterways and Shipping Administration (WSV), communicated by the German Federal Institute of Hydrology
(BfG).





This investigation was partly supported in the context of the joint project EXTREMENESS (Extreme North Sea storm surges and their consequences) funded by the German Federal Ministry of Education and Research (BMBF, Förderkennzeichen [03F0758]).



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




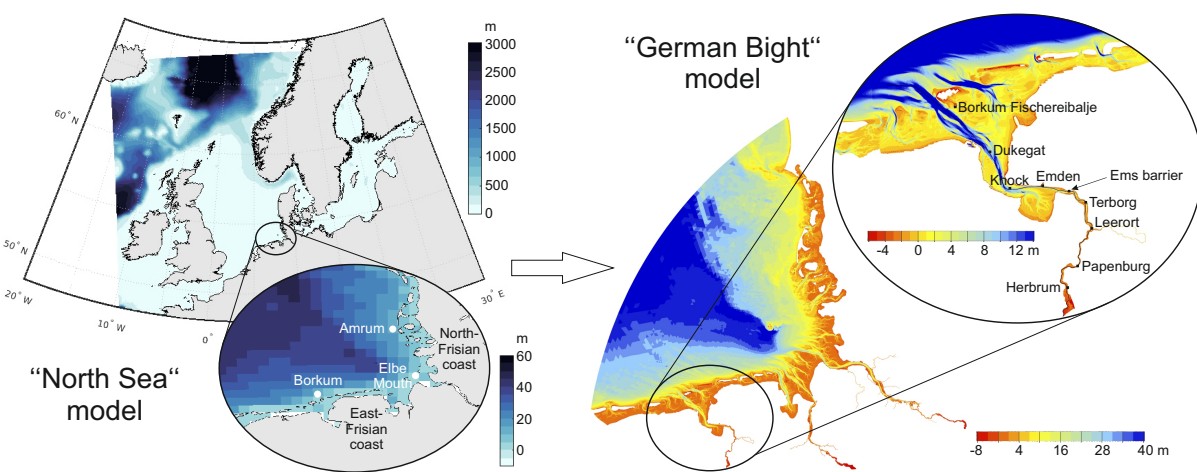

**Figure 1.** Model domains for the North Sea and German Bight models with distributions of water depths and zooms into the German Bight and the Ems estuary, respectively. The black line in the zoom for the Ems estuary denotes the longitudinal profile for which the highest high waters were extracted.

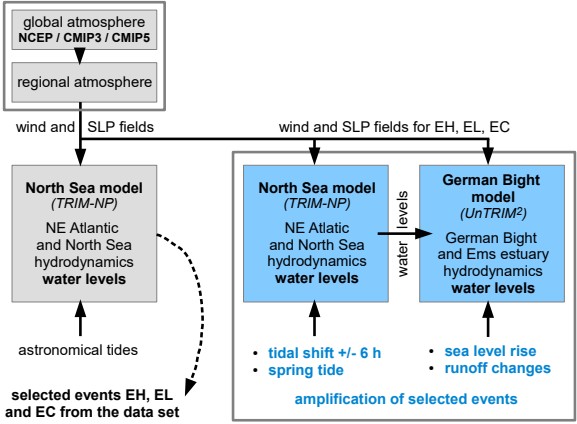

**Figure 2.** Scheme of the chain of models and simulations for the selected events and their amplification.

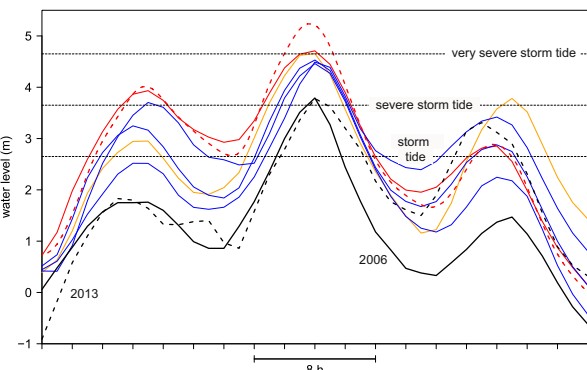

**Figure 3.** Time series of hourly water levels over 36 h for Borkum. The two black curves display observations (Data source: "German Federal Waterways and Shipping Administration (WSV)", communicated by the German Federal Institute of Hydrology (BfG)) and the coloured curves represent the five highest simulated storm tides (nearest seaward grid point to Borkum) from the data set. The red and the orange curves denote the two highest events (EH and EL) used for amplification tests, the dashed red curve displays the amplified event EH_a (see chapter 3.1.2).

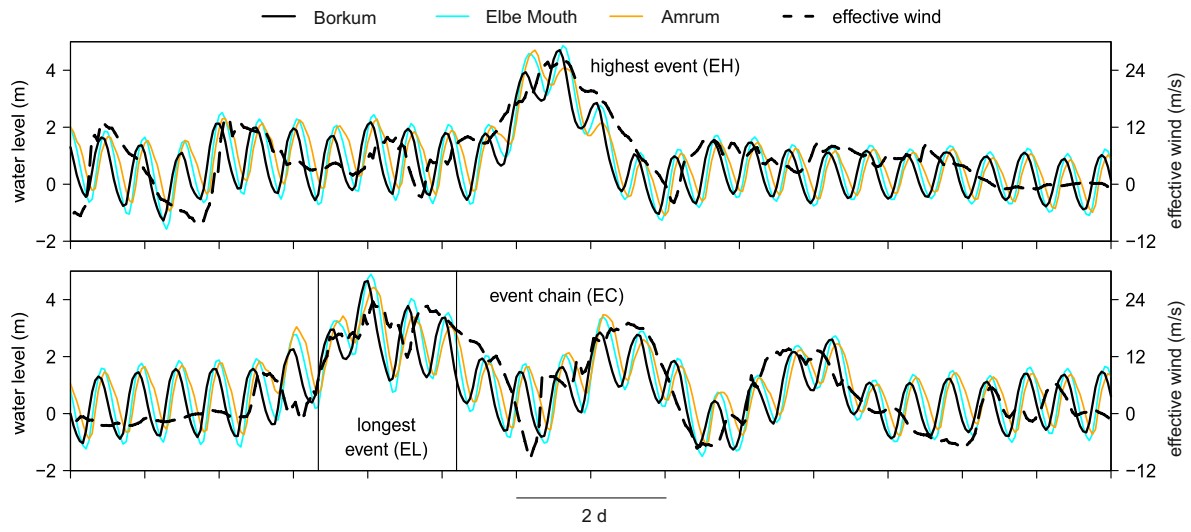

**Figure 4.** Time series of simulated hourly water levels for three locations along the German Bight coast together with effective wind velocities for Borkum (dashed black) for the "highest" event (EH, top) and the "event chain" (EC, bottom) over 14 days. The first event in the event chain represents the "longest" event (EL).

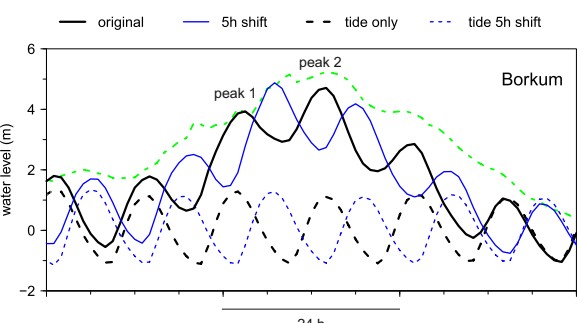

**Figure 5.** Highest event (EH): Time series of the original (solid black line) and amplified water levels (solid blue line) for Borkum together with tide-only time series (dashed black and blue lines), the amplification is due to 5-hourly shifting of the tide. The dashed green curve presents the effective wind from -10 to $30\,\mathrm{ms}^{-1}$.


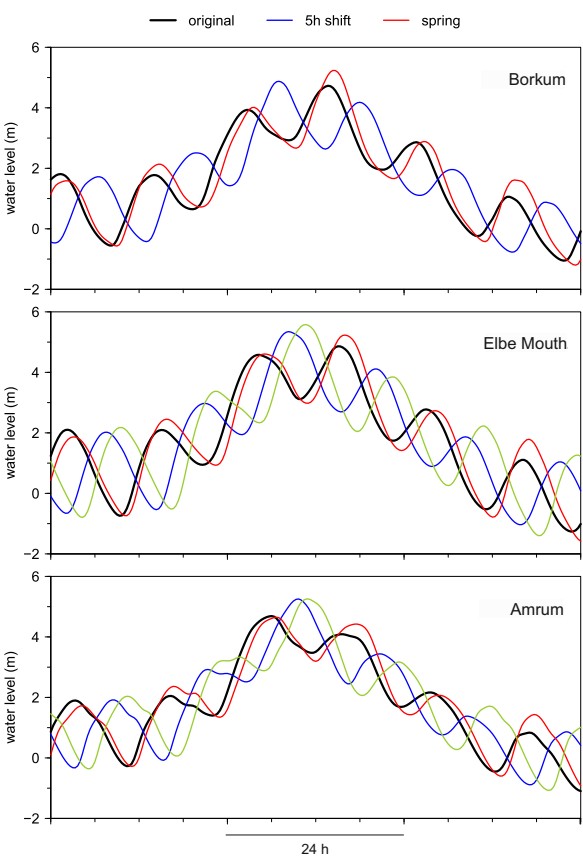

**Figure 6.** Highest event (EH): Time series of the original (black lines) and amplified water levels due to 5-hourly shifting of the tide (blue lines) and due to replacement of the original tide by the highest spring tide together with shifting of the tide (red lines) for three locations along the German Bight coast. The olive curves show water levels with the strongest amplification for Elbe Mouth and Amrum.




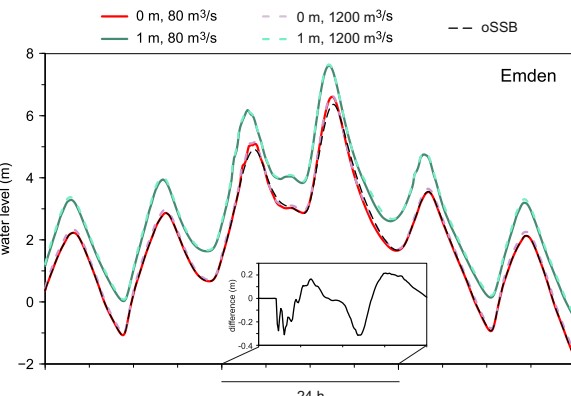

**Figure 7.** Highest event (EH_a): Time series of the amplified water levels for the location Emden in the Ems estuary. The two solid and two dashed curves display the water levels for Q equal $80\,\mathrm{m^3\,s^{-1}}$ and $1200\,\mathrm{m^3\,s^{-1}}$ and a SLR of $0\,\mathrm{m}$ and $1\,\mathrm{m}$, respectively, and operated storm surge barrier. The dashed black curve shows water levels for an open storm surge barrier (oSSB), for $Q=80\,\mathrm{m^3\,s^{-1}}$ and no SLR. The insert shows the differences between the water levels for operated (red curve) and open storm surge barrier (dashed black curve) over 24 h.

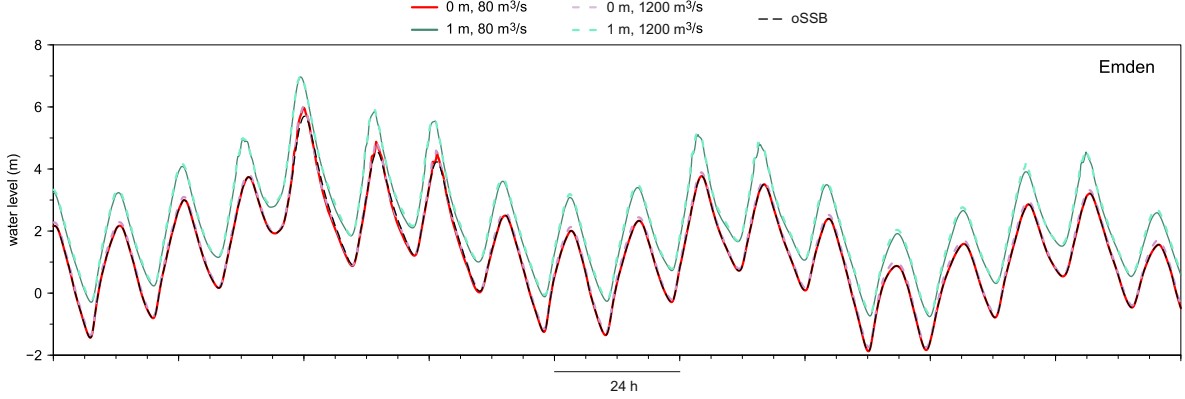

**Figure 8.** Longest event/event chain (EL/EC): Times series of the amplified water levels as in Figure 7.

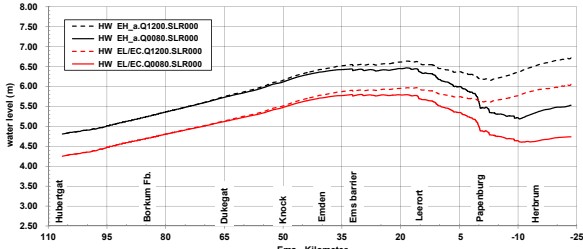

**Figure 9.** Highest water level along a longitudinal profile in the Ems estuary during EH_a (black lines) and EL/EC (red lines) for $Q=80\,\mathrm{m^3\,s^{-1}}$ (solid lines) and $Q=1200\,\mathrm{m^3\,s^{-1}}$ (dashed lines) for an open storm surge barrier without SLR.





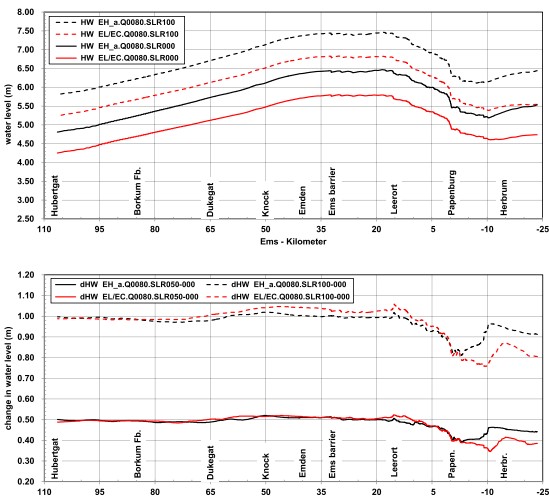

**Figure 10.** top: Highest water level along a longitudinal profile in the Ems estuary during EH_a (black lines) and EL (red lines) for SLR=0 cm (solid lines) and SLR=100 cm (dashed lines) for an open storm surge barrier and Q=80 m$^3$s$^{-1}$. bottom: Differences in the highest water levels between simulations with and without SLR along the Ems estuary during EH_a (black lines) and EL (red lines) for Q=80 m$^3$s$^{-1}$ (dashed lines for SLR=100 cm, solid lines for SLR=50 cm). The storm surge barrier is open.

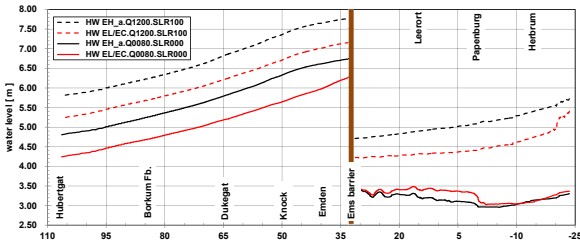

**Figure 11.** Highest water level along a longitudinal profile in the Ems estuary for an operated storm surge barrier without further amplification (Q=80 m$^3$s$^{-1}$, SLR=0 cm; solid lines) and a simulation with enhanced conditions (Q=1200 m$^3$s$^{-1}$, SLR=100 cm; dashed lines) for EH_a (black lines) and EL (red lines).




**Table 1.** Highest water levels and occurence times of highest water levels at Emden are given for simulations with varying river runoff Q, sea level rise SLR and modes of operation for the storm surge barrier. The occurence time of the highest water level is given relative to the Ems-kilometer 107 (Hubertgat).

| event | Q | SLR | barrier | highest water level | occurence time |
|---|---|---|---|---|---|
|  | $[\mathrm{m}^3\mathrm{s}^{-1}]$ | [cm] |  | [m] | [min] |
| EH_a | 80 | 0 | open | 6.36 | 89 |
| EH_a | 1200 | 0 | open | 6.43 | 88 |
| EH_a | 80 | 100 | open | 7.37 | 82 |
| EH_a | 1200 | 100 | open | 7.42 | 79 |
| EH_a | 80 | 0 | operated | 6.61 | 80 |
| EH_a | 1200 | 0 | operated | 6.65 | 74 |
| EH_a | 80 | 100 | operated | 7.61 | 70 |
| EH_a | 1200 | 100 | operated | 7.65 | 69 |
| EL/EC | 80 | 0 | open | 5.71 | 79 |
| EL/EC | 1200 | 0 | open | 5.78 | 77 |
| EL/EC | 80 | 100 | open | 6.75 | 62 |
| EL/EC | 1200 | 100 | open | 6.82 | 61 |
| EL/EC | 80 | 0 | operated | 5.96 | 73 |
| EL/EC | 1200 | 0 | operated | 6.01 | 71 |
| EL/EC | 80 | 100 | operated | 6.96 | 45 |
| EL/EC | 1200 | 100 | operated | 7.01 | 45 |
| 13.03.1906 | 167* | - | no | 5.18* | - |
| 01.11.2006 | 32** | - | operated | 5.17* | - |

\* DGJb (2018)

\*\* personal communication WSA Meppen (2019)





**Table 2.** Duration of water levels higher than a selected limit (here NHN - 1.40 m) and MHW (at Knock MHW = NHN + 1.39 m, at Emden MHW = 1.48 m) for the event EC with varying river runoff Q, sea level rise SLR and operated storm surge barrier. The period investigated covers 176 hours.

| | Knock | | Emden | |
|---|---|---|---|---|
| event | > -1.40m | > MHW | > -1.40m | > MHW |
| | hours | hours | hours | |
| EC.Q0080.SLR000 | 172 | 88 | 172 | 86 |
| EC.Q1200.SLR000 | 176 | 91 | 176 | 91 |
| EC.Q0080.SLR100 | 172 | 129 | 173 | 127 |
| EC.Q1200.SLR100 | 176 | 131 | 176 | 130 |