# Peer review of "Extreme storm tides in the German Bight (North Sea) and their potential for amplification."

_Natural Hazards and Earth System Sciences, 2019_

## Referee Comment (RC1) · Andreas Sterl (Referee) · 18 Dec 2019

**Synopsis**

The paper investigates possible maximum water levels as resulting form storm surges along the German coast, and especially in the Ems estuary. From a large set of simulations (hydrodynamic model driven by atmosphere model output) the highest events are selected. These events are then re-calculated while changing the phase difference between astronomical tide and storm: What if the storm occurred a bit earlier or later during the tidal cycle? It is shown that a different timing of the storm could increase the water level by up to 0.5 m.

In a second, somehow unrelated, part the development of the water level in the Ems estuary is investigated, thereby including/excluding the existing surge barrier, taking river discharge into account, and increasing the mean sea level by 1 m. Not surprisingly, closing the barrier lowers the water level upstream of the barrier and slightly increases it downstream. River discharge has only a small impact as its volume is small compared to the volume of the estuary. A higher sea level (due to climate change) just adds up to the water level.

**Discussion**

"What is the highest possible water level due to a storm?" The paper aims to shed some light on this very important question. The authors do so by just shifting the phase between astronomical tide and storm. As the two are independent, the highest wind speed may occur at any moment during a tidal cycle, so by shifting them with respect to each other, one can investigate possible increases of water level without invoking any new physics or improved models. As a comprehensive hydrologic model is used, the (nonlinear) interaction between astronomical tide and storm surge is automatically accounted for. Although not completely new, this approach is a valuable contribution to our knowledge of high water levels.

**Detailed comments**

**language** The text should be proof-red by a native speaker.

**punctuation** A lot of sentences would be easier to understand if they contained some more commas, e.g., p 3, line 78/79: comma before *was used*; or p 5, line 127: comma before *in particular*.

**title** *Enhancement* means *an increase or improvement in quality, value, or extent.* I do not think that this is meant here. Simply *increase* might be more appropriate.

**p 2, l 29** higher water levels - higher than what?

**p 2, l 39-41** Please reformulate. I do not understand.

**p 4, l 118-120** Are these CORDEX simulations? If so, please mention. I helps the reader to recognize these simulations.

**p 5, l 151** *wind speed maximum* - where is the maximum taken? Does it matter, by the way? As far as I understand, you shift the astronomical tide with respect to the whole storm, don't you?

**p 5, l 141** chain of events - this criterion should be explained in more detail. What does it mean? The maximum number of storms in a week? The longest storm? Successive storms from different directions?

**p 6, bottom** For model performance the reader is referred to earlier publications. Fine, but for most readers one or two general sentences about the model quality would remove the necessity to look up those papers.

**p 7, l 207-208** the height of the gates was increased from 7 to 8 m in nature to 9 m. – I am confused about the height of the gates. Has the height of the real gates already been increased from 7 to 8 m? Or did you increase them twice in the model?

**p 8, l 221** original simulated events – you mean the simulations without shift of lag between tides and storm?

**p 8, l 226** Please specify the EC event – how does it look like?

**p 8, l 246** 15 cm, but previously you mentioned water levels of 3.93 m and 4.88 m, the difference of which is 95 cm. It's a bit confusing. Just reformulate the sentence, and it will become much clearer why the increase is only 15 cm.

**p 9, l 158** by only *a* few centimeters

**p 9, l 164-266** I do not understand what is meant here

**p 11, ll 330** rises → raises

**p 12, l 372** it's the *insert* in Figure 7 that you have to look at

**p 13, l 411-413** hard to follow, please reformulate

**p 14, l 446** last word: there → their

**Fig. 3, caption line 3** the data set – which data set? Caption should be comprehensible without reading the main text

**p 24, Figs. 7 and 8** the dashed lines (0 m, 1200 $m^3$/s, and 1 m, 1200 $m^3$/s) are not visible. Probably, they are covered by the respective solid lines. If so, please mention in the caption.

---

## Referee Comment (RC2) · Anonymous Referee #2 · 16 Feb 2020

The paper presents an interesting analysis of plausible amplifications of storm tides at the German Bight and the Ems estuary. These amplifications are investigated in terms of numerical model simulations that account for changes in storminess from climate change scenarios, changes in the timing of the tidal phase and spring tide conditions for the open coast; in addition to SLR and increase of river discharge for the Ems estuary. Results show that the largest amplification of the storm tidal levels (up to 50cm) at the open coast arises from a shift of the tidal phase with respect to the storm peak. In the Ems estuary, the SLR causes the largest amplifications at the mouth, while its effects are negligible in the inner part of the estuary. The opposite is true for an increase of the river discharge. Finally, the effects of closing the storm surge barrier of the estuary is also investigated. Closing the barrier produces an increase of the

water levels at the outer part of the estuary and a decrease in the inner part. However, the effects in the inner part depend on the river discharge and for how long the barrier is closed.

Although this study is interesting and relevant, I agree with the first reviewer that the manuscript should be proof-red by a native speaker and many commas are missing (I listed some in the minor comments below). In addition, how some sentences are structured and some terminology used makes the reading difficult. Specifically, I recommend to state first what it was done, and then give the reasons why it was done in that way, instead of the opposite order. The paper is long and it addresses several different scenarios and analyses (climate changes scenarios, changes in the tidal phase...), so the use of generic terms in some cases (e.g. L218 "data set") makes difficult the reading because it is not clear to which simulation/data set the authors are referring to. In addition, I would use "event" instead "ensemble member" e.g. L239, L247.

In the introduction and discussion, I am also missing more references of other studies using a similar approach than in this paper as well as other studies made for the German Bight region. For example, Arns et al. (2015) analyzed the non-linear effects of different SLR scenarios on the peaks of storm tides at the German Bight and Santamaria-Aguilar et al. (2017) assessed the effects of these scenarios on the storm surge hydrographs. In addition, Wahl et al. (2011) developed a statistical approach for generating a large number of storm surge events.

I find the section of data, methods and experiments difficult to read and follow due to the large number of datasets, models and simulations made/used. However, the summary and discussion is well structured and clear. I recommend to rephrase first sentence of section 2.2 and to re-structure the section ordering the different data sets e.g. Start with the hydrodynamic model used, hindcast forcing, and climate change scenarios and models. (However, it is very clear in the diagram of Figure 2). In some cases, the reading would be easier if the type/variables is specified e.g. "multi-decadal hindcast"

or "climate realizations", which can refer to atmospheric forcing or water levels. It would also be interesting to know the length of the hindcast and climate scenarios period i.e. specify the years.

Minor comments:

Title: I suggest to change "very severe" for extreme, which is the term commonly used in the literature and actually, it is also used in the manuscript e.g. L138 (Here and along the manuscript). In addition, enhancement can also be changed to "amplification", which is the term used more often along the manuscript.

L1. Change "essential" for major

L20. This sentence is vague.

L25. Environmental threat-> Natural hazard/threat

L27.Inflicted heavy losses-> caused large damages

L29-30. Rephrase. The use of commas is not correct in this sentence.

L30. Risk of what?

L31. Remove associated with anthropogenic climate change.

L32. Storm surge-> If storm tides is the term used, please be consistent along the manuscript.

L32-33. Link this sentence with the previous one. In addition, references can be added as e.g. Arns et al. 2015

L61. . . .forcing, a possible amplification can occur or possible amplifications

L63-64. Add comma after variations and considered.

L65. Comma after study.

L68. The climate realizations used, comprising CMIP3 and CMIP5 scenarios, reflect

only. . ..and local bathymetric changes or changes in the local bathymetry.

L75. Comma after set. Remove distinct

L76. Simulations of what?

L79. Comma after surges

L82. Comma after estuary.

L84. Comma after Emden

L94. Comma after Bight

L99. The Ems estuary is situated in the southern German Bight, at the border. . . (Remove North Sea because the location of the German Bight was already specified).

L136. Rephrase. For instance, "The methodology used to investigate the potential amplification of the storm tide events comprises four steps"

L141. It is not clear here how an event is defined, which is explained in L212-215. These lines should be moved to this section as they are part of the methods and not of the results.

L152. If the SLR is not included in the simulations of climate scenarios of the North Sea model, why the largest spring tide of each climate scenario is used and why it would change between them? Are the climate scenarios for different periods? How is the tide extracted from the simulated water levels?

L153. Comma after two.

L154-155. Rephrase this sentence.

L166. Remove "To the North Sea" and add "ocean" boundary of the German Bight model.

L204. Comma after conservation.

L217. Comma after 3.

L219. Remove comma.

L238. Comma after EH.

L243-245. Divide the sentence in two and add commas.

L251. Comma after EH.

L252. Change "except"-> "with exception of"

L254. Comma after water and members.

L258. How much was the increase? These lines are too vague: "few centimeters", "not a substantial increase", "nearly no changes"....

L262. Rephrase. Single high waters?

L264-266. Move to section 2.5.

L278-279. There is no need of explaining again where Elbe mouth and Amrum are located.

L285. Change differing to different.

L285. Comma after Amrum.

L287. Rephrase: "The olive curves of both Elbe mouth and Amrum correspond to the same simulation, which incorporates both the largest spring tide and the phase shift of the tide".

L289-291. Use duration above MHW instead of time period

L299. Comma after amplification.

L300. Comma after the parenthesis. Is EH_a instead of EH?

Figures 7 & 8. Dashed lines cannot be clearly differentiated. I also recommend to add

a line showing the MHW level in figure 8 as the changes of the duration above this level are discussed.

L318. Was a simulation with a SLR of 0.5m also performed? This was not mentioned before.

L325. The highest

Figures 9, 10 & 11. Font size of legends, axes and labels is too small.

L326. Add parenthesis (HW).

L330. Rises-> raises

L338. Comma after addition. Is decreasing-> decreases.

L350-357. I do not understand these lines and why are in this section. Rephrase them and move them to the discussion. (Or simply remove them, because it is repeated in lines 435-438)

L380. Increases o causes an increase of.

L382. Highest-> High or an increase of the highest

L412. There-> They are from

L413. Clarify this line. The absence of considerable increase of storm surges correspond to the magnitude or frequency? Because this study is focused only on 3 types of events, but it does not include any analysis of changes in the trends/ variability of storm surges.

L433. Rephrase.

LL460. Particular

References:

Arns, A., Wahl, T., Haigh, I. D., and Jensen, J. (2015). Determining return water levels

at ungauged coastal sites: a case study for northern Germany. Ocean Dyn. 65, 539–554. doi:10.1007/s10236-015-0814-1.

Santamaria-Aguilar, S., Arns, A., and Vafeidis, A. T. (2017). Sea-level rise impacts on the temporal and spatial variability of extreme water levels: A case study for St. Peter-Ording, Germany. J. Geophys. Res. Ocean. 122, 2742–2759. doi:10.1002/2016JC012579.

Wahl, T., Mudersbach, C., and Jensen, J. (2011). Assessing the hydrodynamic boundary conditions for risk analyses in coastal areas: a stochastic storm surge model. Nat. Hazards Earth Syst. Sci. 11, 2925–2939. doi:10.5194/nhess-11-2925-2011.

---

## Author Comment (AC1) · 27 Mar 2020

*We would like to thank the reviewer for taking the time to review our manuscript and for the valuable comments and suggestions to improve our manuscript. We respond to the comments by referring to the page and line numbers in the original manuscript.*

title: Enhancement means an increase or improvement in quality, value, or extent. I do not think that this is meant here. Simply increase might be more appropriate.

*We agree and changed ''enhancement'' to ''amplification''*

p 2, l 29 higher water levels - higher than what?

*The sentence is changed:*
*''Mainly due to these measures more recent storms, e.g. 1976 or 2013, caused no severe damages although water levels higher than those of 1962 have been observed at various coastal sections …''*

p 2, l 39-41 Please reformulate. I do not understand.

*The second part of the paragraph is reformulated.*

''This information is usually assessed and provided in form of high percentiles or return values obtained from frequency distribution estimates. There is a spectrum of methods used to construct such estimates (e.g. Debernard and Røed (2008), Arns et al. (2015b), Santamaria-Aguilar et al. (2017) for dynamical modeling approach, Wahl et al. (2011) for stochastic modelling 40 approach or Dangendorf et al. (2013) for processing of tide gauge observations). In the present study we are interested in the spatial and temporal evolution of particular very severe storm tide events in coastal areas and estuaries and, thus, diverge from statistical approach. So far, more detailed information and assessment of particular events that are extremely severe and rare are uncommon. Potential sources of such events comprise historical data as well as modelled data for past, present and future.''

p 4, l 118-120 Are these CORDEX simulations? If so, please mention. I helps the reader to recognize these simulations.

*The used simulations were partially from CORDEX, we added more description in the text.*

*Chapter 2 is rearranged and reformulated according to the suggestions of reviewer 2. The description of the area under investigation (2.1) is followed by the description of the ''North Sea'' (2.2) and the ''German Bight'' (2.3) models used in our investigation and by the description of the data set (2.4). Finally, the selection of events and amplification experiments is specified (2.5). Chapters 2.4 and 2.5 are reformulated as follow:*

[revised manuscript text omitted]

p 5, l 151 wind speed maximum - where is the maximum taken? Does it matter, by the way? As far as I understand, you shift the astronomical tide with respect to the whole storm, don't you?

*You are right, we shift the whole fields and the exact wind maximum is unimportant. But we use the wind maximum near Borkum (added in the text) as a reference because it roughly coincides with water level maximum and helps us to identify the time frame where we are looking for the new water level maximum.*

p 5, l 141 chain of events - this criterion should be explained in more detail. What does it mean? The maximum number of storms in a week? The longest storm? Successive storms from different directions?

*Chain of events is explained in more detail in section 2.5 (former section 2.3), see above.*

p 6, bottom For model performance the reader is referred to earlier publications. Fine, but for most readers one or two general sentences about the model quality would remove the necessity to look up those papers.

*A sentence is added.*
"The model has been validated against tide gauge observations at the German coasts."

p 7, l 207-208 the height of the gates was increased from 7 to 8 m in nature to 9 m. – I am confused about the height of the gates. Has the height of the real gates already been increased from 7 to 8 m? Or did you increase them twice in the model?

*In reality some gates have the height 7 m and some are 8 m. They all were set to 9 m in the model simulation. The sentence is modified to clarify this.*
"…. the height of the gates were increased from 7 m (2 gates) respectively 8 m (5 gates) in nature to 9 m in the model."

p 8, l 221 original simulated events – you mean the simulations without shift of lag between tides and storm?

yes

p 8, l 226 Please specify the EC event – how does it look like?

*Chain of events is explained in more detail in section 2.5 (former section 2.3), see above.*

p 8, l 246 15 cm, but previously you mentioned water levels of 3.93 m and 4.88 m, the difference of which is 95 cm. It's a bit confusing. Just reformulate the sentence, and it will become much clearer why the increase is only 15 cm.

*The sentences are reformulated to clarify the differences.*

''Due to the diurnal inequality, peak 1 of the corresponding astronomical tide is about 20 cm higher than peak 2. Due to the 5 h shifting, peak 1 of the tide coincides with stronger wind velocities, whereas peak 2 coincides with weaker wind velocites. Thus, by only shifting the astronomical tide against the wind field, an amplification of the maximum high water in the event EH of 15 cm (from original 4.73m to 4.88 m) is obtained.''

p 9, l 158 by only a few centimeters
p 9, l 164-266 I do not understand what is meant here

*The text is reformulated.*

''…by only a few centimeters. In the original event EL the highest high water already coincides with an astronomical spring tide about 7 cm lower than the highest one. Thus, both applied procedures lead to relative changes of the three highest water level peaks, however not to a substantial absolute increase of the maximum water level during EL. Furthermore, the length of EL shows nearly no changes. Possible amplification was also tested for the entire EC event including EL. The storm tides following EL experience an increase of some single high waters up to 20 to 30 cm together with a decrease of other high waters for some ensemble members. Thus, there was no general amplification regarding the intensity (see chapter 2.5) of the event chain EL/EC. Therefore, the amplification procedures for EL/EC were discarded.''

p 11, ll 330 rises ! Raises

*changed*

p 12, l 372 it's the insert in Figure 7 that you have to look at

*The sentence is changed accordingly.*

''This effect can be observed e.g. in Figure 7 looking at the water level of the event EH_a (red line) and in the insert of Figure 7 showing the difference between the water levels for operated and open storm surge barrier.''

p 13, l 411-413 hard to follow, please reformulate

*The sentence and the following text are reformulated.*

''These events originate from the first half of the emission scenario period of two different climate realizations. Gaslikova et al. (2013) showed that the annual maximum water levels of these climate realizations displayed strong multi-decadal variability but no significant long-term trends from 1961 to 2100. Thus, the found highest water levels exceeding the water levels measured since the

beginning of the 20th century at Borkum (Figure 3) could be possible already under present-day conditions as no sea level rise is included in the original climate realizations.''

p 14, l 446 last word: there ! Their

*changed*

p 24, Figs. 7 and 8 the dashed lines (0 m, 1200 m3/s, and 1 m, 1200 m3/s) are not visible. Probably, they are covered by the respective solid lines. If so, please mention in the caption.

*We add a sentence that the dashed and solid lines are similar.*
''As the impact of Q on the water levels at Emden is small, the dashed red and green curves nearly match the solid red and green curves.''

---

## Author Comment (AC2) · 27 Mar 2020

*We would like to thank the reviewer for taking the time to review our manuscript and for the valuable comments and suggestions to improve our manuscript. We respond to the comments by referring to the page and line numbers in the original manuscript.*

Although this study is interesting and relevant, I agree with the first reviewer that the manuscript should be proof-red by a native speaker and many commas are missing (I listed some in the minor comments below). In addition, how some sentences are structured and some terminology used makes the reading difficult. Specifically, I recommend to state first what it was done, and then give the reasons why it was done in that way, instead of the opposite order. The paper is long and it addresses several different scenarios and analyses (climate changes scenarios, changes in the tidal phase...), so the use of generic terms in some cases (e.g. L218 "data set") makes difficult the reading because it is not clear to which simulation/data set the authors are referring to. In addition, I would use "event" instead "ensemble member" e.g. L239, L247.

*We went again though the text clarifying when possible and addressing the specific comments of the reviewers. The "data set" definition was explained in the Line 164 or it was explicitly stated in the text when some other data were meant. We think it is important to keep the "ensemble member" term, although we agree that the usage of this term is unconventional in this context. We want to underline the deference between an "event" – a particular atmospheric situation and corresponding surge, which is unique for a given atmospheric situation and an "ensemble member" – a particular constellation of tidal component and surge component and there are many of them for a given atmospheric situation.*

In the introduction and discussion, I am also missing more references of other studies using a similar approach than in this paper as well as other studies made for the German Bight region. For example, Arns et al. (2015) analyzed the non-linear effects of different SLR scenarios on the peaks of storm tides at the German Bight and Santamaria-Aguilar et al. (2017) assessed the effects of these scenarios on the storm surge hydrographs. In addition, Wahl et al. (2011) developed a statistical approach for generating a large number of storm surge events.

*Thank you for pointing out the necessity of additional references also including other methods of analysis. There was a considerable amount of studies during the past decades investigating storm tides in the German Bight, so we wanted to limit references only to the relevant methodology. Now the introduction is partly reformulated and more references are added.*

*"This information is usually assessed and provided in form of high percentiles or return values obtained from frequency distribution estimates. There is a spectrum of methods used to construct such estimates (e.g.Debernard and Røed(2008), Arns et al. (2015b), Santamaria-Aguilar et al. (2017) for dynamical modeling approach, Wahl et al. (2011) for stochastic modeling approach or Dangendorf et al. (2013) for processing of tide gauge observations). In the present study we are interested in the40 spatial and temporal evolution of particular very severe storm tide events in coastal areas and estuaries and, thus, diverge from statistical approach. So far, more detailed information and assessment of particular events that are extremely severe and rare are uncommon. Potential sources of such events comprise historical data as well as modelled data for past, present and future."*

I find the section of data, methods and experiments difficult to read and follow due to the large number of datasets, models and simulations made/used. However, the summary and discussion is well structured and clear. I recommend to rephrase first sentence of section 2.2 and to re-structure the section ordering the different data sets e.g. Start with the hydrodynamic model used, hindcast forcing, and climate change scenarios and models. (However, it is very clear in the diagram of Figure 2). In some cases, the reading would be easier if the type/variables is specified e.g. "multi-decadal hindcast" or "climate realizations", which can refer to atmospheric forcing or water levels. It would also be interesting to know the length of the hindcast and climate scenarios period i.e. specify the years.

*Chapter 2 is rearranged and reformulated according to the suggestions. The description of the area under investigation (2.1) is followed by the description of the ''North Sea'' (2.2) and the ''German Bight'' (2.3) models used in our investigation and by the description of the data set (2.4). Finally, the selection of events and amplification experiments is specified (2.5).*

*Chapters 2.4 and 2.5 are reformulated as follow:*

[revised manuscript text omitted]

Minor comments:
Title: I suggest to change "very severe" for extreme, which is the term commonly used in the literature and actually, it is also used in the manuscript e.g. L138 (Here and along the manuscript). In addition, enhancement can also be changed to "amplification", which is the term used more often along the manuscript.

*"enhancement" is changed to "amplification" and "very severe" to "extreme"*

L1. Change "essential" for major

*changed*

L25. Environmental threat-> Natural hazard/threat

*changed to ''natural hazard''*

L27.Inflicted heavy losses-> caused large damages

*changed*

L29-30. Rephrase. The use of commas is not correct in this sentence.

*The sentence is changed*

*''Mainly due to these measures more recent storms, e.g. 1976 or 2013, caused no severe damages although water levels higher than those of 1962 have been observed at various coastal sections …''*

L30. Risk of what?

*''of flooding'' is inserted*

L31. Remove associated with anthropogenic climate change.

*changed*

L32. Storm surge-> If storm tides is the term used, please be consistent along the manuscript.

*All ''storm surge'' terms are changed into ''storm tide''.*

L32-33. Link this sentence with the previous one. In addition, references can be added as e.g. Arns et al. 2015

*The text is reformulated and the reference Arns et al. 2015 is added.*

*"In modern times, two major storm tide disasters that caused large damages at the North Sea coasts occurred in the years 1953 and 1962. Since then coastal defenses have been significantly improved throughout the coastline. Mainly due to these measures more recent storms, e.g. 1976 or 2013, caused no severe damages although water levels higher than those of 1962 have been observed at various coastal sections (NLWKNa(2010), NLWKNb(2007)). Nevertheless, risk of flooding is still present and may increase due to expected climate change. Thus, the rise of the mean sea level may lead not only to an increase in the height of the storm tides and longer duration of water levels exceeding certain thresholds (e.g. Idier et al. (2019) and references therein) but also to shorter arrival times of the storm tide at the coast and in the estuaries (e.g. Arns et al. (2015a)). These effects, among others, may aggravate risks related to storm tides and may have consequences for coastal protection e.g. for the dike heights or the warning times, but also for such issues as the drainage of low-lying coastal areas."*

L61. : : :forcing, a possible amplification can occur or possible amplifications
L63-64. Add comma after variations and considered.
L65. Comma after study.

*Commas are inserted.*

L68. The climate realizations used, comprising CMIP3 and CMIP5 scenarios, reflect only.....and local bathymetric changes or changes in the local bathymetry.
L75. Comma after set. Remove distinct

*changed*

L76. Simulations of what?

*''Water level'' is inserted.*

L79. Comma after surges
L82. Comma after estuary.
L84. Comma after Emden
L94. Comma after Bight

*Commas are inserted.*

L99. The Ems estuary is situated in the southern German Bight, at the border....
(Remove North Sea because the location of the German Bight was already specified).

*changed*

L136. Rephrase. For instance, "The methodology used to investigate the potential amplification of the storm tide events comprises four steps"

*changed*

L141. It is not clear here how an event is defined, which is explained in L212-215. These lines should be moved to this section as they are part of the methods and not of the results.

*Chapter 2 is rearranged and reformulated according to the suggestions. Events are explained more precisely and these lines are now at the beginning of chapter 2.5 ''Selection of events and amplification experiments'', see above.*

L152. If the SLR is not included in the simulations of climate scenarios of the North Sea model, why the largest spring tide of each climate scenario is used and why it would change between them? Are the climate scenarios for different periods? How is the tide extracted from the simulated water levels?

*The procedure description is reformulated.*

*'' For ensemble two, the highest astronomical spring tide found in the tidal simulations for the period 1948-2100 was used instead of the original tide and the astronomical tides were shifted again hourly.''*

L153. Comma after two.

*inserted*

L154-155. Rephrase this sentence.

*This sentence is reformulated. See Chapter 2.5 above.*

L166. Remove "To the North Sea" and add "ocean" boundary of the German Bight model.

*Changed according to the suggestions.*

L204. Comma after conservation.
L217. Comma after 3.
L219. Remove comma.
L238. Comma after EH.

*All changed.*

L243-245. Divide the sentence in two and add commas.

*changed*

L251. Comma after EH.
L252. Change "except"-> "with exception of"
L254. Comma after water and members.

*All changed.*

L258. How much was the increase? These lines are too vague: "few centimeters", "not a substantial increase", "nearly no changes"......
L262. Rephrase. Single high waters?

*The text is changed and values are added.*

*"In case of the longest event EL (included in EC Figure 4), both amplification procedures - shifting of the astronomical tide against the wind and replacement of the original astronomical tide with the highest spring tide together with shifting - result in an increase of the highest high water by only a few centimeters. In the original event EL the highest high water already coincides with an astronomical spring tide about 7 cm lower than the highest one. Thus, both applied procedures lead to relative changes of the three highest water level peaks, however not to a substantial absolute increase of the maximum water level during EL. Furthermore, the length of EL shows nearly no changes. Possible amplification was also tested for the entire EC event including EL. The storm tides following EL experience an increase of some single high waters up to 20 to 30 cm together with a decrease of other high waters for some ensemble members. Thus, there was no general amplification regarding the intensity (see chapter 2.5) of the event chain EL/EC. Therefore, the amplification procedures for EL/EC were discarded."*

L264-266. Move to section 2.5.

*The text is slightly changed, but we think it is useful at this place to clarify again which events are transferred to be simulated with the German Bight model.*

L278-279. There is no need of explaining again where Elbe mouth and Amrum are located.

*The phrase is removed.*

L285. Change differing to different.
L285. Comma after Amrum.

*changed*

L287. Rephrase: "The olive curves of both Elbe mouth and Amrum correspond to the same simulation, which incorporates both the largest spring tide and the phase shift of the tide".

*The sentence is changed*

*''The olive curves of both Elbe Mouth and Amrum correspond to the same ensemble member, which incorporates both the largest spring tide and a phase shift of the tide.''*

L289-291. Use duration above MHW instead of time period
L299. Comma after amplification.
L300. Comma after the parenthesis. Is EH_a instead of EH?

*All changed.*

Figures 7 & 8. Dashed lines cannot be clearly differentiated. I also recommend to add a line showing the MHW level in figure 8 as the changes of the duration above this level are discussed.

*A line showing MHW is include in Figure 8 and a sentence concerning the similarity of the dashed and the solid lines is added.*

L318. Was a simulation with a SLR of 0.5m also performed? This was not mentioned before.

*It was mentioned on L165*

L325. The highest
Figures 9, 10 & 11. Font size of legends, axes and labels is too small.
L326. Add parenthesis (HW).
L330. Rises-> raises
L338. Comma after addition. Is decreasing-> decreases.

*All changed*

L350-357. I do not understand these lines and why are in this section. Rephrase them and move them to the discussion. (Or simply remove them, because it is repeated in lines 435-438)

*We added the reason for the analysis to clarify the aim of the investigation.*
*Here, we address the point, that a chain of events is not only important with respect to coastal protection but also for the drainage of the low lying hinterlands. We explain, which water level in one sewer exemplarily is important for draining and we use this threshold in table 2. We think this text is useful here. In the discussion it is only mentioned that EC would hinder natural drainage.*

"During storm tides not only questions concerning coastal protection are important, but the draining of the protected areas during storm tides must be ensured, too. In the lowlands close to the mouth of the Ems draining of urban (e.g. Emden) and agricultural areas (e.g. Knock) is of major interest. The aim of the sewer at Knock is to drain the low lying hinterland (with a ground level of about NHN + 0 m) and keep the inland water level at Knock lower than NHN - 1.40 m (KLEVER (2018)). At Knock the mean low water MLW is NHN - 1.58 m so that draining without pumping is only possible for a short time even during mean tides. Caused by long lasting high water levels during storm tides draining is even more restricted. For the chain of storm tides EC (Figure 8) even without amplification pumping is needed nearly during the whole period of 176 hours (Table 2). The water must be pumped against a water level in the Ems higher than MHW for about 90 hours. This period will increase by about 40 hours in case of a sea level rise of 100 cm."

L380. Increases o causes an increase of.
L382. Highest-> High or an increase of the highest
L412. There-> They are from

*All changed*

L413. Clarify this line. The absence of considerable increase of storm surges correspond to the magnitude or frequency? Because this study is focused only on 3 types of events, but it does not include any analysis of changes in the trends/ variability of storm surges.

*That is correct, we do not investigate long-term trends in this study, however, the used met-ocean data sets were analyzed earlier. The text is changed accordingly.*

''These events originate from the first half of the emission scenario period of two different climate realizations. Gaslikova et al. (2013) showed that the annual maximum water levels of these climate realizations displayed strong multi-decadal variability but no significant long-term trends from 1961 to 2100. Thus, the found highest water levels exceeding the water levels measured since the beginning of the 20$^{th}$ century at Borkum (Figure 3) could be possible already under present-day conditions as no sea level rise is included in the original realizations.''

L433. Rephrase.

*The sentence is changed:*

''Against the background of climate change and the need to develop future coastal protection strategies it is not only important to know the possible height of an extreme event but also its duration.''

L460. Particular

*changed*

---

## Referee Report (RR1)

The manuscript "Extreme storm tides in the German Bight (North Sea) and their potential for amplification" by Grabemann et al. analyses if the the most extreme storm events in the German Bight could become more severe under slightly different conditions using hydodynamic model simulations. With a high resolution model of the Ems estuary they additionally analyse comound flod events through river runoff and a storm surge. All in all this is a very interesting impact study. The manuscript is mostly written in a clear and concise manner but should be revised by a native speaker.

Therefore I do recommend this manuscript for publication.

Minor concerns:

p 2, l 26 what kind of coastal defenses? An example would be helpful.

p 2, l 29 "due to the [effects of] expected climate change"

p 2, l 31 rise of mean sea level will also effect the nonlinear interactions between tide and surge (see also Arns et al. 2015, 2020)

p 3, l 68 but bathymetric changes or an influence of mean sea level rise on the morphodynamics is very likely so this is a big Problem, that this still can not be accounted for. Would be nice if they can give a hint that this research question is still unanswered.

p 3, l 77 is it a 2-D model? Please be specific. So the reader can be sure that no baroclinic effects are accounted for.

p 4, l 89 Study area, data and methods

p 4, l 103 "… German [reference height system] standard …"

p 4, l 115 wind speed and direction. Please be specific.

p 5, l 137 it is sad that there was no better solution for the salinity input. I mean the approch is fine to use (especially in the North Sea, where the influence should be very small), but there is room for improvement.

p 5, l 148 maybe better: total water level

p 6, l 171 introducing the abbreviation "mean tidal high water [(MHW)]"

---

## Author Response (AR2)

*We would like to thank the reviewer for taking the time to review our manuscript and for the valuable comments and suggestions to improve our manuscript. We respond to the comments by referring to the page and line numbers in the manuscript.*

Minor concerns:

p 2, l 26 what kind of coastal defenses? An example would be helpful.

*"Since then coastal defenses have been significantly improved throughout the German coastline. For example, the main dykes were reinforced, secondary dykes were introduced and storm surge barriers were constructed to protect coasts of adjoined rivers."*

p 2, l 29 "due to the [effects of] expected climate change"

*changed*

p 2, l 31 rise of mean sea level will also effect the nonlinear interactions between tide and surge (see also Arns et al. 2015, 2020)

*We agree, the nonlinear interactions between tide and surge will be influenced by the SLR. However here we are rather interested in the entire storm tide events than in the identification and assessment of different components. The nonlinearity itself is considered by the hydrodynamic modeling.*

p 3, l 68 but bathymetric changes or an influence of mean sea level rise on the morphodynamics is very likely so this is a big Problem, that this still can not be accounted for. Would be nice if they can give a hint that this research question is still unanswered.

*We agree that bathymetric changes and morphodynamics are an important issue. We address the point that we used a fixed bathymetry and neglect bathymetric changes due to natural processes and anthropogenic influences in section 4, lines 465-478. Some changes were made there. In this study, we focus on the physically plausible storm tides related in the first place to variability of the atmospheric forcing.*

*"Consideration of a changing bathymetry would give an insight on the effect of morphodynamic states on extreme storm tides. Due to the lack of regular information about the past and possible future regional and local changes, this remains an important and extensive topic for further investigation."*

p 3, l 77 is it a 2-D model? Please be specific. So the reader can be sure that no baroclinic effects are accounted for.

*The models are described in details further in the text (sections 2.2, 2.3), we included the clarification here as well.*
*"a regional hydrodynamic model in a 2-D mode"*

p 4, l 89 Study area, data and methods

*changed*

p 4, l 103 "… German [reference height system] standard …"

*changed*
*"NHN (Normalhöhennull) presents the standard elevation zero of the German reference height system"*

p 4, l 115 wind speed and direction. Please be specific.

*changed*

p 5, l 137 it is sad that there was no better solution for the salinity input. I mean the approch is fine to use (especially in the North Sea, where the influence should be very small), but there is room for improvement.

*You are right, the salinity may be implemented in a more sophisticated way. However, it was beyond the scope of this study to develop plausible scenarios for the salinity variations.*

p 6, l 171 introducing the abbreviation "mean tidal high water [(MHW)]"

*The abbreviation was introduced in Section 2.1, line 102.*

[revised manuscript text omitted]